# Development and Evaluation of a Workplace Bullying Cognitive Rehearsal-Based Nursing Simulation Education Program: A Mixed-Methods Study

**DOI:** 10.3390/ijerph20064974

**Published:** 2023-03-11

**Authors:** Mijeong Park, Jeong Sil Choi

**Affiliations:** 1Department of Nursing, Hoseo University, Asan-si 31499, Republic of Korea; 2College of Nursing, Gachon University, Incheon 21936, Republic of Korea

**Keywords:** cognitive rehearsal, coping skills, nursing education, simulation training, workplace bullying

## Abstract

Bullying makes learning difficult for nursing students, and using real-life scenarios during training can improve the understanding of workplace bullying. Thus, to mitigate bullying experienced by nurses, this study developed and evaluated a cognitive rehearsal education program that constituted training nursing students through role-play simulations. A mixed-method research design was used to evaluate 39 nursing students from two universities. A quasi-experimental research design was applied to assess symptoms, knowledge, and perceptions of bullying, and a focus group interview was conducted with six participants. Quantitative analyses revealed that the program improved participants’ knowledge and perceptions but not their symptoms. The focus group interview revealed that the program improved participants’ coping skills and desire for education. This program could be effective in raising awareness of workplace bullying and increasing relevant coping skills. It can be further developed as part of an overall strategy to manage workplace bullying and its consequences in hospital settings.

## 1. Introduction

Workplace bullying refers to mental, verbal, or physical attacks conducted amongst coworkers in professional settings, exacerbated by the risks of scarce support and social exclusion for those attacked [1]. Psychological threats are more common than physical threats [2], greatly affecting employees’ quality of work, which is extremely important in workplaces such as hospitals. Leymann [1] used the term “mobbing” to describe serious interpersonal conflict in the workplace and named “bullying” for the first time in the United Kingdom for a similar phenomenon [3]. Other terms often cited to describe bullying behaviors include mobbing, relational aggression, horizontal violence, and harassment [3,4]. “Mobbing” limits perpetrators to a group; however, “bullying” is a higher-level concept that extends the concept of “mobbing” and includes bullying by one person [2,3].

According to the literature, the common nature of workplace bullying includes the negative and unwanted nature of the behavior, repetition and persistence, and imbalance of power [1,2,3]. A negative act includes repetitive and persistent behaviors that last for more than 6 or 12 months and are experienced once a week or more [1,2]. An imbalance of power exists between the victim and perpetrator, and it is difficult for the victim to protect themselves from the bullying. Furthermore, it tends to be more severe due to reduced coping skills and increased feelings of helplessness [3].

Traditionally, individual and organizational factors have been investigated as antecedents of bullying [1,3,4]. Individual factors were related to general and sociopsychological characteristics and self-esteem [3,4]. Some studies reported that age was a significant factor and that new nurses were more likely to become victims of workplace bullying than experienced nurses [4,5]. In a systematic review, low self-esteem was an individual antecedent of workplace bullying among nurses [5]. The personalities of both the perpetrator and the victim may be involved as causes of both bullying behavior and perceptions of being bullied. At the organizational level, leadership, work design, and the morale of management and the workforce have been investigated as major causes of bullying [1,2,4]. Compared with other occupations, nurses have a particularly serious problem with workplace bullying. The causes mainly include workload and tension in an urgent hospital environment, job stress, role conflict, and low autonomy in an organization in which the hierarchy of power is obvious [6]. In a hospital environment with a strong hierarchical order and high work stress, nurses are in a weak position regarding power imbalance. In addition, workplace bullying often occurs due to the characteristics of these nursing organizations [7]. Therefore, to prevent bullying, strategic approaches at the individual and organizational levels are required depending on the nature and cause.

Workplace bullying is an important issue in nursing education and hospital management, and its incidence rates vary worldwide depending on countries’ laws and organizational characteristics [8]. In a systematic review, the pooled prevalence of workplace bullying among Korean nurses was 22.2% [5]. Yet, bullying among nurses and nursing students has existed for such a long time that it may be considered a preparatory experience [9,10]. It is difficult for nurses to recognize bullying and take effective and immediate action against it [11]; moreover, they also experience psychological symptoms such as depression and anxiety and physical symptoms such as weight loss and fatigue [12].

In nursing education, preparing for the fight against bullying requires a systematic approach that takes into account the specificities of university education and clinical practice [5,6,7]. Effective nursing interventions to reduce bullying should be adjusted according to the developmental stage of nursing students [13]. Through a scoping review, methods of nursing and school interventions to reduce bullying were classified into various categories [14]. Prevention programs that were applied to schools and proved to be effective included a social cognitive theory-based intervention (reduced bullying and victimization) [15], bullying prevention program (improved social skill) [16], and pragmatic school-based universal intervention (improved mental health) [17]. Therefore, it is necessary to provide nursing students with interventions that have been proven effective in preventing bullying. Furthermore, it is necessary to consider mastering the correct techniques for dealing with bullying along with an accurate understanding of its causes and mechanisms [18,19,20]. In addition, education climates directly affect students. Faculty, clinical administrators, and educators should create an environment in which bullying is not taken for granted and must take the lead in supporting students to overcome bullying experiences in a constructive and positive manner [8,10,11].

Bullying also makes learning difficult for nursing students, but it is tricky to teach them effective coping methods [21]. Among the educational methods used in nursing, simulation-based learning is the most effective, as it induces active participation, presents realistic clinical settings, and enables repeated practice and experience [21]. It also enhances learners’ competencies and can demonstrate action-oriented outcomes [21]. Therefore, it can improve awareness of workplace bullying and develop coping strategies for the same. So far, workplace bullying prevention programs include the cognitive rehearsal program by Griffin [22], in which new nurses rehearsed coping skills in bullying situations, the transition program for new nurses [23], and assertiveness training [24]. Cognitive rehearsal intervention is a strategy for negotiating stressful or emotional situations and is based on cognitive behavioral therapy [22,23]. Its core factors include knowledge and practices, perceptions and recognizing bullying behavior, managing personal reactions, rehearsing responses in a collaborative and safe environment, and confidence and increasing skill [22,23]. However, the effectiveness of these programs has remained inconsistent, and very few interventions have targeted nursing students in their critical phases of learning and practice [25,26].

Hence, this study aimed to develop a cognitive rehearsal-based simulation program to prevent workplace bullying among nursing students. The program provided simulations of clinical settings, qualitatively evaluated bullying experiences through in-depth interviews, and analyzed effects on symptom experiences, knowledge, and perceptions of workplace bullying.

## 2. Materials and Methods

### 2.1. Design

A concurrent embedded mixed-method research design was applied, including a focus group interview and a quasi-experimental research design involving a one-group pretest–post-test design [27] (Figure 1).

### 2.2. Participants

Participants included fourth-year students with at least one year of clinical practice and no previous exposure to simulation education or classes in workplace bullying at four-year nursing colleges in cities A and I in Korea. Using G*Power 3.1.9.7 software (http://www.gpower.hhu.de/ (accessed on 28 June 2020), with a significance level of 0.05, power of 80%, and effect size of 0.5, a minimum sample size of 34 was required. Considering drop-outs, 50 participants were recruited; data from 39 who underwent the entire experimental process were analyzed, while responses from 11 were excluded due to incompletion. Data for the qualitative analysis were collected through a focus group interview with six participants who voluntarily agreed to participate in the interview and were among those whose data were included in the final analysis.

### 2.3. Developing the Cognitive Rehearsal-Based Nursing Simulation Education Program

The program was developed using the ADDIE framework (Analysis, Design, Development, Implementation, and Evaluation) based on Jeffries’s [28] simulation model.

#### 2.3.1. Demand Analysis

To analyze their needs, individual interviews were conducted with 12 fourth-year nursing students who had undergone clinical practice. Results revealed high demand for coping skills, followed by understanding the meaning and causes of workplace bullying, respectively.

#### 2.3.2. Program Design

The program follows the order of “Scenario running → Reflection → Debriefing,” based on the role-play bullying simulation by Gillespie et al. [21].

#### 2.3.3. Development

Based on the demand analysis and the programs by Griffin [23] and Stagg et al. [29], a scenario module was developed. The scenario involved nursing a patient with a herniated disc, during which a senior nurse would bully a junior nurse, with a trained standard patient (student majoring in theater and film), a trained senior nurse, and three or four nursing students. It depicted three work-related, four personal, and three physically intimidating cases of bullying. The total duration was 90 min, including 5 min for orientation and dispensation, 20 min of practice, 20 min for scenario analysis and reflection, and 45 min of debriefing (Table 1).

Four experts assessed the program’s validity in terms of its goals, content and composition, operating time, and method appropriateness, after which the simulations were revised and supplemented. The range of content validity coefficient was 0.75–1.00 when calculated for each item based on a 4-point scale. Subsequently, a pilot study was conducted with five junior nurses to identify and correct potential issues in operating the program. Consequently, sufficient opportunities to share comments and ask questions during debriefing were added. The final goals of this program were to enable participants to:Explain the meaning and types of workplace bullying;Accurately recognize workplace bullying;Effectively respond to workplace bullying;Utilize related resources and ask for help when required;Demonstrate appropriate coping skills.

#### 2.3.4. Testing

Before implementation, the scenario was practiced with an assistant researcher three times, and the program was simulated once to enhance realism and recognize any unexpected variables in advance.

### 2.4. Evaluation of the Program

The demand survey was conducted from 1 July to 5 July 2020, and the program development period was one month. The effectiveness of the program was evaluated through quantitative and qualitative analyses between 13 August and 20 September 2020.

#### 2.4.1. Preliminary Survey

A questionnaire was administered two weeks before the program to measure participants’ characteristics, symptom experiences, and knowledge and perceptions of workplace bullying.

#### 2.4.2. Scenario Implementation Stage

Participants were grouped into 13 teams of three to four students. In the scenario running stage, the trained standard patient and trained senior nurse conducted the simulation. In the reflection stage, the participants completed a short questionnaire with open questions assessing self-reflections on bullying and 5-point Likert scale items to verify whether they immediately recognized bullying. In the debriefing stage, the definition and types of bullying, as well as coping skills, were discussed.

#### 2.4.3. Quantitative Evaluation of Final Effects

A postsurvey questionnaire was administered two weeks after the program to measure symptom experiences and knowledge and perceptions of workplace bullying.

#### 2.4.4. Qualitative Evaluation of Program Participation

To explore the longevity of changes caused by the program, a focus group interview lasting between 90 and 100 min was conducted with six participants two weeks after the program. With prior consent, the interview was recorded and transcribed. Important questions included, “How did this program affect your perception of workplace bullying?” “Are you confident that you can apply what was learned in real life?” “Are there foreseeable difficulties when applying what was learned today in real life?” “What do you think needs to be added to resolve workplace bullying in nursing?”

### 2.5. Measures

#### 2.5.1. Symptom Experience

Symptom experience was measured using the Brief Symptom Inventory-18 developed by Derogatis [30], which was validated by Park et al. [31] in Korean. It has 18 questions across four factors: five questions on somatization, six on depression, four on anxiety, and three on phobic anxiety. Participants indicated their daily experiences of each symptom on a 5-point Likert scale, in which 1 = “not at all” and 5 = “very severe.” Higher scores indicate higher experiences of negative symptoms. In the study by Park et al. [31], Cronbach’s α was = 0.89, and in this study, it was 0.91.

#### 2.5.2. Knowledge of Workplace Bullying

The tool for measuring knowledge of workplace bullying was developed by the investigator using the Negative Acts Questionnaire-Revised [32] and the Assessment of Knowledge in Bullying developed for medical students [33]. It consists of 16 questions in total, selected through a preliminary survey with six nursing students, whose content validity was confirmed by two head nurses, one charge nurse, and one professor of nursing. The 16 questions included 12 on personal bullying, 7 on work-related bullying, and 3 on physically intimidating bullying, with higher scores indicating higher levels of knowledge. Correct answers were assigned a score of 1, while incorrect or empty responses received 0. The total score was converted based on a score of 100. The final content validity was = 0.95; the Kuder–Richardson 20 was 0.65 in the preliminary survey and 0.72 in the actual survey.

#### 2.5.3. Perceptions of Workplace Bullying

This tool was developed by the investigator with five questions based on the tool by Quetta, who surveyed bullying perceptions among medical students [33]. The content validity was assessed by four experts. Items included, “I understand the definition of workplace bullying,” “I understand the negative effects of workplace bullying,” “I recognize cases of workplace bullying,” and “I recognize the coping skills for workplace bullying,” rated on a 5-point Likert scale with 1 = “not at all” and 5 = “very much so.” Higher scores indicate higher levels of perception. In a preliminary survey conducted with six nursing students, Cronbach’s α was = 0.89, whereas in the actual study, it was 0.91.

### 2.6. Analysis

Quantitative data were analyzed using SPSS/WIN 23.0 (IBM Corp, Armonk, NY, USA), with *p* = 0.05. The effects of the program were analyzed using the nonparametric Wilcoxon signed-rank test (Shapiro–Wilk test). Qualitative data from the focus group interview were analyzed using the content analysis method. After transcribing the recorded interviews, multiple reviews were conducted to extract key concepts, phrases, and words. Relationships among extracted concepts were categorized under abstract themes, which were comprehensively analyzed to derive the final themes.

### 2.7. Ethics

This study was conducted after obtaining approval from the Institutional Review Board (IRB NO. 1044396-201910-HR-183-01) of G University prior to data collection. Participants were informed that the data collected would be handled anonymously, would not be used for purposes other than this study, and that they could refuse participation in the study at any time. Written consent was obtained from each participant.

## 3. Results

### 3.1. Participants’ Characteristics

Participants included 34 women (87.2%) and 5 men (12.8%) with a mean age of 22.6 years, with 21 (53.8%) participants aged 21 or 22. Participants had an average of 18 months of clinical practice experience, while none of them had received education on workplace bullying (Table 2).

### 3.2. Effects of the Program

#### 3.2.1. Immediate Recognition of Workplace Bullying by Self-Reflection

Immediate recognition of workplace bullying in the scenario learning stage was 2.75 ± 0.99 points out of 5, with recognition of personal bullying being the highest with 2.91 ± 1.09 points. Among the 10 specific bullying cases included in the scenario, “excessive monitoring” scored the highest with 3.48 ± 1.16 points, and “alienating” was the lowest with 2.48 ± 1.40 points.

Participants indicated psychological abuse, anger, neglect, and discrimination as types of bullying and expected difficulties such as excessive stress, psychological pain, low self-esteem, and social phobia. In addition, their action responses were passive, such as no response, endurance, or silence, whereas some reported notifying superiors and devising solutions after checking relevant information (Table 3).

#### 3.2.2. Evaluating Final Effects

Prior to the program, scores for total symptom experience were 1.82 ± 0.83 out of 5; after the program, scores were 1.64 ± 0.67, not showing significant differences. However, in the subcategories, scores for depression were significantly different (*Z* = 2.029, *p* = 0.049).

Scores for knowledge increased from 69.23 ± 7.77 out of 100 to 83.33 ± 17.10, showing significant differences (*Z* = −5.522, *p* =< 0.001). Similarly, scores for perceptions increased from 3.14 ± 0.39 out of 5 to 3.66 ± 0.46, showing significant differences (*Z* = −2.436, *p* = 0.020). Among detailed items, there were significant differences in knowing the meaning of workplace bullying and perception of coping skills (Table 4).

### 3.3. Exploring Participation Experiences

Two topics and six subtopics regarding participation were derived from the qualitative focus group interview.

1:Improved coping ability

Participants indicated that the program enabled them to understand the meaning of workplace bullying, accurately recognize bullying situations, and learn coping skills such as utilizing related resources and requesting help. In addition, they indicated that their fears of workplace bullying were reduced due to the program.

1a:Ability to accurately recognize workplace bullying

Participants reported that they were able to accurately recognize and describe workplace bullying situations.


*After participating in the program, I realized that I only had a theoretical idea of workplace bullying. I think I would have been confused about bullying if it had occurred in a real clinical setting … Now, I feel like I can accurately understand and explain which behaviors constitute workplace bullying.*
(Participant 2)

In addition to conventionally known examples of bullying, such as personal attacks, shouting, and abusive language, participants reported that they were able to recognize subtler instances, such as not receiving patient information, being ordered to do work below their qualifications, or being given unrealistic goals.


*When I was a newbie, I thought that I had to do the things that senior nurses generally don’t like to do, such as assistant jobs or errands, until I got better because I was just learning. And I thought I should not feel bad about these things … If I had not learned in this program, I would not have known that being made to do things that which do not like to do is bullying.*
(Participant 3)

1b:Learned coping skills

Participants indicated that they learned effective coping skills against workplace bullying.


*If I had not learned in this training, I would not have known what to do and felt helpless when bullied. However, now I think I can do what I have learned if I feel bullied.*
(Participant 5)

Participants also mentioned that they learned of antibullying manuals at work, immediate recommended measures, and related laws and gained confidence that they could take more active stances against workplace bullying.


*I thought that I should not notify the head nurse unless I planned to resign. If there is an antibullying manual at the ward and if everyone knows about it, I plan to notify the head nurse without hesitation and with no fear of revenge when bullying occurs.*
(Participant 4)

1c:Reduced fear of workplace bullying

Participants expressed a fear of workplace bullying, which reduced after the program.


*I heard about the high turnover rates for nurses due to the bullying culture at work. I often worried about becoming a victim of bullying, how much would I be able to endure, what if I quit my job at the hospital because of bullying and so on … Now that I clearly understand workplace bullying and know what to do, I am not so afraid.*
(Participant 2)

2:Need for continuous education

Participants reported improved hopes of dealing with workplace bullying but were still worried about whether they could effectively cope in various clinical settings. Thus, continuous and repetitive education is required.

2a:Wanting to respond effectively to various workplace bullying situations

Participants felt limited in effectively responding to workplace bullying situations and reported requiring more education for the same.


*I mean, who knows which hospital and which department I would be assigned to work at after graduation? I want to go to the emergency room or the ICU, but I heard that the nurses working there are even worse and so is bullying because everyone is so busy. A lot of nurses quit. I want to receive more training in bullying situations that can occur in the emergency room or the ICU so I can adapt well when I am stationed there.*
(Participant 1)

Participants also wished to learn coping skills against bullying by doctors or other professionals, not just nurses.


*I heard that doctors tend to look down on nurses. They say that work becomes difficult if you do not get along with doctors. I would like to know what to do in situations like that.*
(Participant 3)

2b:Recognizing the possibility of becoming the bully rather than the bullied

Participants reminded themselves of individual responsibilities in efforts to prevent workplace bullying and recognized the fact that they could be the bullies rather than the bullied.


*If I had become a senior nurse without properly learning about workplace bullying, I think I would have done exactly as the bullies who bullied me, without any guilt, under the name of training new nurses … To do away with the bullying culture at the hospital, I think everyone should receive this education mandatorily.*
(Participant 4)

2c:Wanting to receive continuous and repetitive coping training

Participants mentioned that they might forget key learnings and requested continuous and repeated education.


*I wonder if I could remember what I’ve learned now when I actually experience bullying. I wish there were opportunities to repeat similar situations in clinical practice or in-class practice. If not, I wish there was a book available at the department that I could go back to for reference.*
(Participant 3)

## 4. Discussion

This cognitive rehearsal program demonstrated effectiveness in developing participants’ understanding of and coping skills for workplace bullying [29,34]. The novelty of this study lies in incorporating a simulation training method to imitate real-life bullying scenarios, after which participants’ abilities to recognize and respond to bullying were quantitatively and qualitatively assessed. The rehearsal method, a traditional intervention method corroborated to be effective, has shown positive effects on nurses’ knowledge and awareness of bullying [22,29,35] but not on their symptom experiences [36].

In this study, the cognitive rehearsal simulation significantly improved knowledge, as demonstrated in previous studies [22,26,29]. Importantly, what should be noted is that the cognitive aspect of applying knowledge was enhanced by this program. However, the knowledge score in the preliminary survey was 69.23 points out of 100, but the postsurvey score for immediate recognition of workplace bullying was low at 2.75 points out of 5. This indicates that nursing students have theoretical knowledge of bullying but are unable to recognize it during actual experiences. Hence, education should extend beyond simply delivering information; instead, experiences of real-life situations should be provided.

Self-reflection and debriefing enabled effects to be maximized among participants, similar to previous studies that have applied the role-play simulation method [37]. Additionally, knowledge scores were maintained at a high of 83.33 points out of 100 during the postsurvey, meeting previous recommendations for evaluating knowledge as a program effect [37]. Such changes in knowledge are not due to simple memorization but due to self-reflection and experience, which are expected to impact behavioral outcomes more than other interventions.

Self-reflection and correction of perceptions during debriefing improved participants’ perceptions, as has been found in previous studies [22,26,29]. Significant improvements were observed in the understanding of and coping with bullying, probably because this program focused on clarifying and training participants on the definition, types, and coping skills for workplace bullying. Accurately recognizing cases of bullying is the first step toward coping; hence, case-based and repetitive education should be provided.

As workplace bullying affects psychological symptoms [12], the interventions’ effects on improving symptom experiences must be evaluated. In this program, no significant improvements were found for somatization, anxiety, and phobic anxiety, and only depression showed a significant decrease. This occurred possibly due to exogenous effects caused by the 4-week interval between the pre- and postsurveys. Additionally, some similar studies did not report any effects on symptom experience [36]. Such psychological symptoms affect not only nurses’ well-being but also occupational burnout and turnover intention [38,39,40]. Therefore, improving psychological symptoms is extremely important, and future studies should control variables that can affect symptom experiences. Moreover, qualitative findings indicate that bullying education can cultivate healthy nursing work environments by reducing the fears associated with bullying and improving confidence in responding to bullying. Moreover, the participants expressed the desire to receive further training in developing coping skills in different environments. For instance, it is important to provide bullying education and training in clinical settings to both reduce the likelihood of developing bullying behaviors and increase coping [12,21]. It is also necessary to share experiences by linking the results of experience identification with behaviors of violence through these programs.

### Limitations

One limitation of this study is the usage of a single-group pre- and post-test design. We propose that future studies use a randomized, controlled pre- and post-test design. Second, it would be necessary to include students from various nationalities and practice situations and conduct additional surveys for symptom experience that reduce confounding effects. Lastly, longitudinal studies of burnout, clinical practice outcomes, and actual bullying incidences are also required [25,26].

## 5. Conclusions

The bullying experienced by nursing students in clinical practice has existed for a long time. However, it has been tolerated. In addition, it is difficult to recognize bullying and take effective and immediate action against bullying. Workplace bullying in clinical nursing education requires a systematic approach according to the developmental stage of nursing students while considering individual and organizational characteristics. This study developed, implemented, and evaluated a cognitive rehearsal-based simulation education program to develop bullying awareness and coping skills among nursing students. A mixed-method research design was applied, which included a focus group interview and a quasi-experimental research design. Quantitative analyses revealed that the role-play simulation program improved participants’ knowledge and perceptions but not their symptoms. Regarding the results of the qualitative assessments, nurse students experienced positive feedback on workplace bullying perceptions, their fear disappeared, and their confidence in coping with bullying improved. These results provide a basis for assessing educational needs and effectiveness against bullying in future studies or education programs. In addition, if a cognitive rehearsal-based simulation education program that allows nursing students to continuously experience various situations of workplace bullying in the nursing curriculum is developed, it will contribute to the development of practical nursing education that can prevent and manage workplace bullying.

## Figures and Tables

**Figure 1 ijerph-20-04974-f001:**
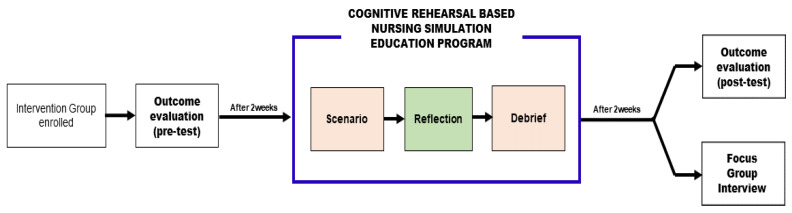
Study design.

**Table 1 ijerph-20-04974-t001:** Demonstration scenarios for the program.

Topic of the Program	Recognizing and Coping with Workplace Bullying while Nursing a Patient with a Herniated Disc
Overview of scenario	Total 90 min in duration - 5 min for orientation and scenario distribution - 20 min for acting out the scenario - 20 min for analysis and reflection on the scenario - 45 min for debriefing Learners: nursing students
Instructions on scenario (5 min for orientation and scenario distribution)	
Case summary (situation)	You are a nurse working in an orthopedic ward. Patient Kim (male, 55 years old) visited the hospital due to severe back pain after falling from his bed and was transferred to the orthopedic ward yesterday after being diagnosed with L4-5 herniation based on emergency room CT results. Patient is complaining of persistent back pain and is not moving at all. Currently, there is no IV route, and the attending physician has issued an order for tarasyn 1ample IM for the pain. Apply appropriate nursing procedures for the patient.
2. Role assignment—scenario casting	(1) Patient (student majoring in theater and film): 1 trained standard patient (2) Senior nurse (nurse): 1 trained senior nurse (3) Junior nurse (nursing student): 3 or 4 nursing students form a group for practice
Scenario (20 min of rehearsal)	
Bullying behavior: work-related bullying	
Not providing patient information	Situation: Rounding for patient identification. Patient complains of severe back pain. “Do you know who this patient is? Identify the patient, although I doubt you can.”
2. Setting unrealistic goals	“What should you do for this patient? I’m not sure if you know how to do anything …”
3. Ordering to do something below qualification	“If you don’t know anything about what to do, just be a caregiver.”
Bullying behavior: personal bullying	
4. Spreading negative rumors	Situation: An intramuscular injection must be performed for pain control. “Do you know how to give an intramuscular injection? I heard that you don’t know how to do things properly.”
5. Alienating	“(Pointing to a particular junior nurse) Nurse OOO, you only watch from now on. You will only cause problems for the patient.”
6. Shouting	Shouting loudly, “Did you explain it properly to the patient?”
7. Excessive monitoring	“Report to me every minute after intramuscular injection. I have to keep an eye on you to make sure you’re doing things right.”
Bullying behavior: physically intimidating bullying	
8. Finger-pointing	“(Pointing to the name tag with a finger) Nurse OOO! Are you sure you identified the patient correctly?”
9. Dismissal from duties	“(Pushing away with hand) Stop what you’re doing. You have too many problems.”
10. Threatening to penalize or give a bad performance evaluation	“If this is how you’re going to work, you’re going to get a bad evaluation score.”
Reflection (20 min)	
Scenario analysis	Identify whether bullying was recognized in the scenario
Self-reflection	(1) Behaviors perceived as bullying (2) Expected difficulties when bullying occurs (3) Action plans in case of bullying
Debriefing (45 min)	
Debriefing composition	Theoretical concepts of workplace bullying: definition, other terms, prevalence of workplace bullying Common bullying behaviors Consequences of bullying: personal, organizational, professional What to do about bullying: reporting, addressing organization-specific policies Cognitive rehearsal intervention: discussion of the cognitive rehearsal technique, responses to common bullying behaviors

**Table 2 ijerph-20-04974-t002:** Participants’ characteristics (*N* = 39).

Characteristic	Category	n (%)
Gender	Female Male	34 (87.2) 5 (12.8)
Age (years)	21–22	21 (53.8)
	23–24	15 (38.5)
	≧25	3 (7.7)
	Mean ± SD	22.6 ± 1.9
Clinical experience (months)	Mean ± SD	18.0 ± 0.0
Education on workplace bullying	Yes No	0 (0.0) 39 (100.0)

SD, standard deviation.

**Table 3 ijerph-20-04974-t003:** Immediate recognition of workplace bullying and self-reflection (*N* = 39).

Immediate Recognition of Workplace Bullying		
Variable (range)		Mean ± SD
Immediate recognition of workplace bullying by self-reflection in the cognitive rehearsal-based nursing simulation education program (1–5)		2.75 ± 0.99
Work-related bullying		2.72 ± 0.99
Not providing patient information		2.59 ± 1.09
2. Ordering to do something below qualification		2.69 ± 1.19
3. Setting unrealistic goals		2.87 ± 1.78
Personal bullying		2.91 ± 1.09
4. Spreading negative rumors		2.57 ± 1.53
5. Alienating		2.48 ± 1.40
6. Shouting		3.17 ± 1.36
7. Excessive monitoring		3.48 ± 1.16
Physically intimidating bullying		2.56 ± 1.13
8. Finger-pointing		2.52 ± 1.18
9. Dismissal from duties		2.61 ± 1.27
10. Threatening to penalize		2.54 ± 1.40
Self-reflection	Answers	
Things perceived as types of bullying	Psychological abuse, shouting, abusive language, physical/verbal violence, anger, isolation, alienation, neglect, discrimination, accusation, acts of contempt, etc.	
Expected difficulties when bullying occurs	Excessive stress, career/aptitude concerns, low self-esteem, low job satisfaction, depression, anxiety, fear, sleep disorder, poor coping ability, psychological pain, atrophy, loneliness, deterioration in health, suicidal ideation, social phobia, mental weakness, job change, resignation, etc.	
Action plans in case of bullying	None, no idea, endurance, silence, notifying superiors, devising solutions after checking relevant information, make efforts to be recognized by the bully, consulting with peers, transfer or resign, ignore, report, relieve stress in other ways, etc.	

SD, standard deviation.

**Table 4 ijerph-20-04974-t004:** Comparing symptoms, knowledge, and perceptions of workplace bullying before and after the program (*N* = 39).

Variable (Range)	Before Mean ± SD	After Mean ± SD	Z *	*p*
Symptom experience (1–5)	1.82 ± 0.83	1.64 ± 0.67	1.881	0.068
Somatization	1.50 ± 0.68	1.42 ± 0.59	0.916	0.366
Depression	2.14 ± 1.10	1.88 ± 0.80	2.029	0.049
Anxiety	2.08 ± 0.99	1.88 ± 0.86	1.780	0.083
Phobic anxiety	1.58 ± 0.85	1.38 ± 0.72	1.519	0.137
Knowledge (0–100)	69.23 ± 7.77	83.33 ± 17.10	−5.522	<0.001
Perception of workplace bullying (1–5)	3.14 ± 0.39	3.36 ± 0.46	−2.436	0.020
Understands the definition of workplace bullying	2.92 ± 0.58	3.38 ± 0.63	−3.815	<0.001
Understands the cause of workplace bullying	3.05 ± 0.65	3.31 ± 0.73	−1.819	0.077
Understands the negative effects of workplace bullying	3.38 ± 0.59	3.54 ± 0.64	−1.233	0.225
Recognizes cases of workplace bullying	3.59 ± 0.60	3.74 ± 0.64	−0.488	0.628
Recognizes the coping skills for workplace bullying	2.77 ± 0.63	2.85 ± 0.84	−2.436	0.020

SD, standard deviation; * Wilcoxon signed-rank test.

## Data Availability

Data sharing not applicable.

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
