# Peer review of "Development and Evaluation of a Workplace Bullying Cognitive Rehearsal-Based Nursing Simulation Education Program: A Mixed-Methods Study"

_ijerph, 2023, doi:10.3390/ijerph20064974_

Round 1

Reviewer 1 Report

Thank you for the opportunity to read this manuscript.

It is clear that the authors paid considerable attention to its development. The entire methodological side of the manuscript is processed precisely and requires high appreciation.

In an effort to help authors increase the scientific quality of their manuscript and meet the demands of a Journal listed in the Web of Science and Scopus databases, the following recommendations can be formulated:

- Although the section of Introduction describes the significance of the researched topic in current literature, a much deeper analysis of the opinions of various authors on the nature of bullying, a deeper investigation of the causes and different approaches to defining bullying, socio-psychological factors of bullying would be requested.

- A separate section should even be devoted to the analysis of approaches in the fight against bullying, and educational approaches and methods of educating students in the field of preparedness for bullying, understanding the mechanisms of its functioning, mastering the correct techniques for dealing with bullying, etc.

- The conclusion of the manuscript should summarize the basic positives and original contributions of the entire study in view of the development of contemporary science in this area.

- It could also hopefully contain at least key recommendations for the management of educational institutions and the enrichment of study curricula.

Good luck!

Author Response

1. Although the section of Introduction describes the significance of the researched topic in current literature, a much deeper analysis of the opinions of various authors on the nature of bullying, a deeper investigation of the causes and different approaches to defining bullying, socio-psychological factors of bullying would be requested. -> 

Thank you for your valuable insights and suggestions. We have revised the sentences in the Introduction as follows:

The nature of bullying involves a persistent and systematic victimization of a colleague or subordinate through the repeated use of various kinds of aggressive behaviors over a long period of time [1,2]. Traditionally, individual and organizational factors have been investigated as antecedents of bullying [1,2]. On an individual level, there are socio-psychological factors, particularly the personalities of both the perpetrator and the victim, which may be involved as causes of both bullying behavior and perceptions of being bullied, respectively. At the organizational level, leadership, work design, and the morale of management and workforce have been investigated as major causes of bullying [1,2]. Therefore, in order to prevent bullying, a strategic approach at the individual and organizational level is required depending on the cause.

2. A separate section should even be devoted to the analysis of approaches in the fight against bullying, and educational approaches and methods of educating students in the field of preparedness for bullying, understanding the mechanisms of its functioning, mastering the correct techniques for dealing with bullying, etc.  --> 

We have added the following text in the Introduction, and added a new reference:

In nursing education, preparing for the fight against bullying requires a systematic approach that takes into account the specificities of university education and clinical practice [5,6,7]. Through a scoping review, methods of nursing and school interventions to reduce bullying have been classified into various categories. There are four types of nursing interventions that can be provided to reduce the incidence of bullying and its impact, namely prevention programs, activities programs, peer group programs, and resilience programs [9].

Prevention programs include social cognitive theory-based interventions and activities such as film clips, group discussions, and exercises. There are also pragmatic school-based programs based on universal intervention education addressing internal factors that cause bullying, including cooperation/communication, empathy, goals/aspirations, problem-solving, self-awareness, and self-efficacy [9].

9. Yosep, I.; Hikmat, R.; Mardhiyah, A.; Hazmi, H.; Hernawaty, T. Method of nursing interventions to reduce the incidence of bullying and its impact on students in school: a scoping review. Healthcare (Basel) 2022, 10(10), 1835. https://doi.org/10.3390/healthcare10101835

3. The conclusion of the manuscript should summarize the basic positives and original contributions of the entire study in view of the development of contemporary science in this area. - It could also hopefully contain at least key recommendations for the management of educational institutions and the  enrichment of study curricula. --> 

Thank you for your valuable comment. We have revised and added the following text to the Conclusions section:

Workplace bullying in clinical nursing education is different from the bullying experienced in general university. It is necessary to systematically and practically provide education regarding workplace bullying within nursing settings. This study developed, implemented, and evaluated a cognitive rehearsal-based simulation education program to develop bullying awareness and coping skills among nursing students. The program improved participants’ knowledge, understanding, and coping abilities related to bullying, and can be further expanded to reduce the instances of workplace bullying and improve nursing students’ professional experiences. In the nursing education curriculum, the expanded application of a cognitive rehearsal-based simulation education program should be established as an essential component for nursing students.

Reviewer 2 Report

The study lies in the incorporation of a simulation training method to mimic real-life bullying scenarios, after which participants' abilities to recognise and respond to bullying were quantitatively and qualitatively assessed.

The procedure is impeccably followed and both quantitative (level of learning and level of impact of the new bullying knowledge) and qualitative (degree to which they believe they would be able to apply it in real life, the margin of a simulation situation like the one they face) assessments are very useful, I believe that here the students' contributions are very interesting and valuable for the identification of more subtle violent behaviours.
It is also interesting to measure the symptomatic experience to analyse the degree of veracity, of real life experience of situations that students may be going through.
it is interesting that it makes it possible to identify in real life what they have learned from theoretical forging.
Validated and perfectly applied and analysed tools.
Perfect for the identification of behaviours, it would have been internal to explore others not introduced in the questionnaire (as there is an interview and qualitative methodology) that would help in this sense.
The study claims to have contributed to improving students' coping skills from understanding what it means to be bullied in the workplace, accurately recognising bullying situations, knowing how they make you feel and the implications they can have on your work efficiency and performance, as well as on patient care and the importance of learning coping skills, such as using related resources and asking for help. In short, denouncing these abusive situations, which are unfortunately more frequent than we think.

Interesting contributions regarding when they may engage in this type of violent behaviour.
Interesting reflections on how easy it can be, in situations of stress, lack of coping skills or difficult situations, to engage in violent behaviour.
The programme succeeds in raising awareness of the problem, which already makes it advisable to publish it. It demonstrates that simulation, as well as working on manual skills and abilities (know-how), can be used for knowing how to be and knowing how to be.

I would like to recommend a greater focus on qualitative results linked to the identification of experiences and behaviours of violence, as it would help to observe and eradicate them in other clinical settings and I understand that they are not always the same as those suffered by professionals. 

Author Response

I would like to recommend a greater focus on qualitative results linked to the identification of experiences and behaviours of violence, as it would help to observe and eradicate them in other clinical settings and I understand that they are not always the same as those suffered by professionals. ->

Thank you for your valuable insights and suggestions.

The qualitative results have been highlighted in the discussion section. We have also added the following in the Discussion and Conclusions sections:

“Moreover, qualitative findings indicate that bullying education can cultivate healthy nursing work environments by reducing the fears associated with bullying and improving the confidence in responding to bullying. Moreover, the participants expressed the desire to receive further training in developing coping skills in different environments. For instance, it is important to provide bullying education and training in clinical settings, to both reduce the likelihood of developing bullying behaviors and increase coping [7,9]. It is also necessary to share experiences by linking the results of experience identification with behaviors of violence through these programs.”

“Workplace bullying in clinical nursing education is different from the bullying experienced in general university. It is necessary to systematically and practically provide education regarding workplace bullying within nursing settings. This study developed, implemented, and evaluated a cognitive rehearsal-based simulation education program to develop bullying awareness and coping skills among nursing students. The program improved participants’ knowledge, understanding, and coping abilities related to bullying, and can be further expanded to reduce the instances of workplace bullying and improve nursing students’ professional experiences. In the nursing education curriculum, the expanded application of a cognitive rehearsal-based simulation education program should be established as an essential component for nursing students.”

Round 2

Reviewer 1 Report

I believe that the authors tried to fulfill the suggestions that were pointed out in the previous review. They tried to deepen the Introduction section, but only elaborated on quoted ideas and opinions that were already mentioned in the section earlier. Unfortunately, the addition of a single source to the list of used literature cannot be considered as a sufficient improvement of the scientific analysis of the theoretical base in the researched area.

Also, despite the addition of 3 sentences to the Conclusion section, it cannot be considered fulfilled to a sufficient extent.

Good luck.

Author Response

I believe that the authors tried to fulfill the suggestions that were pointed out in the previous review. They tried to deepen the Introduction section, but only elaborated on quoted ideas and opinions that were already mentioned in the section earlier.  Unfortunately, the addition of a single source to the list of used literature cannot be considered as a sufficient improvement of the scientific analysis of the theoretical base in the researched area.

-->>Thank you for your comment. We have tried to deepen the Introduction section. We have added five additional literature reviews, the nature of bullying, a deeper investigation of the causes and different approaches to define bullying, and the socio-psychological factors of bullying.

We have also added a separate section in the Introduction through eight additional literature reviews, the analysis of approaches in the fight against bullying, and educational approaches and methods of educating students regarding preparedness for bullying, and mastering the correct techniques to deal with bullying.

To understand the functional mechanisms of bullying, simulation-based learning and a social cognitive theory-based intervention have been also described.

--> We have revised the sentences in the Introduction as follows:

Leymann [1] used the term “mobbing” to describe serious interpersonal conflict in the workplace, and named “bullying” for the first time in the United Kingdom for a similar phenomenon [3]. Other terms often cited to describe bullying behaviors include mobbing, horizontal violence, relational aggression, and harassment [3,4]. “Mobbing” limits perpetrators to a group; however, “bullying” is a higher level concept that extends the concept of “mobbing” and includes bullying by one person [2,3]. According to literature, the common nature of workplace bullying includes the negative and unwanted nature of the behavior, repetition and persistence, and imbalance of power [1–3]. A negative act includes persistent and repetitive behaviors that lasts for more than 6 or 12 months and are experienced once a week or more [1,2]. An imbalance of power exists between the victim and perpetrator, and it is difficult for the victim to protect themselves from the bullying. Furthermore, it tends to be more severe due to reduced coping skills and increased feelings of helplessness [3]. Traditionally, individual and organizational factors have been investigated as antecedents of bullying [1,3, 4]. Individual factors were related to general and socio-psychological characteristics and self-esteem [3, 4]. Some studies reported that age was a significant factor and that new nurses were more to become victims of workplace bullying likely than experienced nurses [4,5]. In a systematic review, low self-esteem was an individual antecedent of workplace bullying among nurses [5]. The personalities of both the perpetrator and the victim, which may be involved as causes of both bullying behavior and perceptions of being bullied. At the organizational level, leadership, work design, and the morale of management and workforce have been investigated as major causes of bullying [1,2,4].

Compared to other occupations, nurses have a particularly serious problem with workplace bullying. The causes mainly included work load and tension in an urgent hospital environment, job stress, and role conflict and low autonomy in an organization where the hierarchy of power was obvious [6]. In a hospital environment with strong hierarchical order and high work stress, nurses were in a weak position regarding power imbalance. In addition, workplace bullying often occurred due to the characteristics of these nursing organizations [7]. Therefore, to prevent bullying, strategic approaches at the individual and organizational levels are required depending on the nature and cause.

3. Einarsen, S.; Raknes, B.I.; Matthiesen, S. Bullying and harassment at work and their relationships to work environment quality: An exploratory study. Euro J Work Organizational Psychol 1994, 4(4),381–401. https://doi.org/10.1080/13594329408410497

4. Kim, Y.; Choi, J.S. Individual and organizational factors influencing workplace cyberbullying of nurses: A cross-sectional study. Nurs Health Sci 2021, 23(3), 715–722. https://doi.org/10.1111/nhs.12858

5. Kang, J.; Lee, M. Pooled prevalence of workplace bullying in nursing: Systematic review and meta-analysis. J Korean Crit Care Nurs 2016, 9(1), 51–65. (In Korean)

6. Roberts, S.; DeMarco, R.F.; Griffin, M. The effect of oppressed group behavior on the culture of the nursing workplace: A review of evidence and interventions for change. J Nurs Manag 2009, 17(3), 288–293.

7. Baillien, E.; De Cuyper, N.; De Witte, H. Job autonomy and workload as antecedents of workplace bullying: A two-wave test of Karasek's Job Demand Control Model for targets and perpetrators. J Occup Organizational Psychol 2011, 84(1), 191–208.

-> We have added a separate section in the Introduction, and eight new references.

 In nursing education, preparing for the fight against bullying requires a systematic approach that takes into account the specificities of university education and clinical practice [9,10,11]. Effective nursing interventions to reduce bullying should be adjusted according to the developmental stage of nursing students [13]. Through a scoping review, methods of nursing and school interventions to reduce bullying were classified into various categories [14]. Prevention programs that were applied to schools and proved to be effective included a social cognitive theory-based intervention (reduced bullying and victimization) [15], bullying prevention program (improved social skill) [16], and pragmatic school-based universal intervention (improved mental health) [17]. Therefore, it is necessary to provide nursing students with interventions that have been proven effective in preventing bullying. Furthermore, it is necessary to consider mastering the correct techniques for dealing with bullying along with an accurate understanding of its causes and mechanisms [18–20]. In addition, education climates directly affects students. Faculty, clinical administrators, and educators should create an environment in which bullying is not taken for granted and must take the lead in supporting students to overcome bullying experiences in a positive and constructive manner [8,10,11].

 Cognitive rehearsal is a strategy for negotiating emotional or stressful situations and is based in cognitive behavioral therapy [22,23]. Its core elements include knowledge and best practices, personal perceptions and recognizing bullying behavior, managing personal reactions, rehearsing responses in a safe and collaborative environment, and increasing skill and confidence [22,23]. However, the effectiveness of these programs has remained inconsistent, and very few interventions have targeted nursing students in their critical phases of learning and practice [25,26].

 13. Selkie, E.M.; Fales, J.L.; Moreno, M.A. Cyberbullying prevalence among US middle and high school–aged adolescents: A systematic review and quality assessment. J Adolesc Health 2016, 58(2), 125–133. https://doi.org/10.1016/j.jadohealth.2015.09.026

14. Yosep, I.; Hikmat, R.; Mardhiyah, A.; Hazmi, H.; Hernawaty, T. Method of nursing interventions to reduce the incidence of bullying and its impact on students in school: A scoping review. Healthcare (Basel) 2022, 10(10), 1835. https://doi.org/10.3390/healthcare10101835

15. Salimi, N.; Karimi-Shahanjarini, A.; Rezapur-Shahkolai, F.; Hamzeh, B.; Roshanaei, G.; Babamiri, M. The effect of an anti-bullying intervention on male students’ bullying-victimization behaviors and social competence: A randomized controlled trial in deprived urban areas. J Res Health Sci 2019, 19(4), e00461

16. Axford, N.; Bjornstad, G.; Clarkson, S.; Ukoumunne, O.C.; Wrigley, Z.; Matthews, J.; Berry, V.; Hutchings, J. The effectiveness of the KiVa bullying prevention program in Wales, UK: Results from a pragmatic cluster randomized controlled trial. Prev Sci 2020, 21(5), 615–626. https://doi.org/10.1007/s11121-020-01103-9

17. Dray, J.; Bowman, J.; Campbell, E.; Freund, M.; Hodder, R.; Wolfenden, L.; Richards, J.; Leane, C.; Green, S.; Lecathelinais, C.; et al. Effectiveness of a pragmatic school-based universal intervention targeting student resilience protective factors in reducing mental health problems in adolescents. J Adolesc 2017, 57, 74–89. https://doi.org/10.1016/j.adolescence.2017.03.009

18. Nguyen, A.J.; Bradshaw, C.; Townsend, L.; Gross, A.L.; Bass, J. A latent class approach to understanding patterns of peer victimization in four low-resource settings. Int J Adolesc Med Health 2016, 32(1), 202–213. https://doi.org/10.1515/ijamh-2016-0086

19. Nguyen, A.J.; Bradshaw, C.; Townsend, L.; Bass, J. Prevalence and correlates of bullying victimization in four low-resource countries. J Interpers Violence 2020, 35(19–20), 3767–3790. https://doi.org/10.1177/0886260517709799

20. Pandey, A.R.; Neupane, T.; Chalise, B.; Shrestha, N.; Chaudhary, S.; Dhungana, R.R.; Bista, B. Factors associated with physical and sexual violence among school-going adolescents in Nepal: Findings from global school-based student health survey. PLoS ONE 2021, 16(3), e0248566. https://doi.org/10.1371/journal.pone.0248566

->We have revised and added the following text to the Conclusion:

The bullying experienced by nursing students in clinical practice has existed for a long time. However, it must be tolerated. In addition, it is difficult to recognize bullying and take effective and immediate action against bullying. Workplace bullying in clinical nursing education requires a systematic approach according to the developmental stage of nursing students, while considering individual and organizational characteristics. This study developed, implemented, and evaluated a cognitive rehearsal-based simulation education program to develop bullying awareness and coping skills among nursing students. A mixed-method research design was applied, which included a focus group interview and a quasi-experimental research design. Quantitative analyses revealed that the role-play simulations program improved participants’ knowledge and perceptions, but not their symptoms. Regarding the results of the qualitative assessments, nurse students experienced positive feedback on workplace bullying perceptions, their fear disappeared, and their confidence in coping with bullying improved. These results provide a basis for assessing educational needs and effectiveness against bullying in future studies or education programs. In addition, if a cognitive rehearsal-based simulation education program that allows nursing students to continuously experience various situations of workplace bullying in the nursing curriculum is developed, it will contribute to the development of practical nursing education that can prevent and manage workplace bullying. 
